# Person–environment fit and medical professionals' job satisfaction, turnover intention, and professional efficacy: A cross-sectional study in Shanghai

Yuyin Xiao[1,2☯], Minye Dong[1,2☯], Chenshu Shi[2], Wu Zeng[3], Zhenyi Shao[4], Hua Xie[4]*, Guohong Li [1,2]*

1 School of Public Health, Shanghai JiaoTong University School of Medicine, Shanghai, China, 2 Center for HTA, China Hospital Development Institute, Shanghai JiaoTong University, Shanghai, China, 3 Department of International Health, School of Nursing & Health Studies, Georgetown University, Washington, DC, United States of America, 4 Centre for Health Statistics and Information, Shanghai Health Commission, Shanghai, China

☯ These authors contributed equally to this work.
* guohongli@sjtu.edu.cn (GL); 13901893401@139.con (HX)

**Data Availability Statement:** All relevant data are within the paper and its Supporting Information files.

## Abstract

### Objectives

Using the person-environment (PE) fit theory, this study aims to explore factors affecting medical professionals' job satisfaction, turnover intention, and professional efficacy, and to examine individual characters associated with PE fit.

### Design and methods

This study used data from the sixth National Health Service Survey conducted in 2018, with a focus on job outcomes among medical professionals in Shanghai. The reliability and validity of the tools for measuring PE and job outcomes were calculated. A structural equation model was used to examine the relationship among person-job (PJ) fit and person-group (PG) fit, job satisfaction, turnover intention, and professional efficacy. Finally, a hierarchical regression model was used to analyze the association between demographic variables and the PJ and PG fit.

### Results

PG fit was directly and positively associated with job satisfaction and professional efficacy. PJ fit had a direct and positive association with job satisfaction but had a direct and negative association with turnover intention. The indirect association of PJ fit with turnover intention was statistically significant. The results from the hierarchical regression analysis showed that younger physicians generally had a lower level of PJ fit and older physicians with higher education tended to have a lower level of PG fit.

**Funding:** This research was supported by Major Key Research Projects in Philosophy and Social Sciences of the Ministry of Education of the People's Republic of China grant no. 18JZD044, and National Natural Science Foundation of China grant no. 72074147.The funders had no role in study design, data collection and analysis, decision to publish, or preparation of the manuscript.

**Competing interests:** The authors have declared that no competing interests exist.

## Conclusions

Medical professionals with higher PJ or PG fit have higher job satisfaction, and those with higher PG fit have higher professional efficacy. The impact of PJ fit on turnover intention was mediated by job satisfaction. Healthcare managers should take actions to effectively promote medical professionals' PJ and PG fit to improve their retention and efficiency.

## Background

Matching employees with their work environment is one of the most widely studied topics in the field of organizational behavior [1]. Person–environment (PE) fit is broadly defined as the compatibility between individuals and their work environment, which occurs when their characteristics match well [2]. PE fit studies are generally focused on four dimensions of the work environment, including person–superior (PS) fit, person-organization (PO) fit, person-job (PJ) fit, and person-group (PG) fit [1, 3].

It is important to understand the level of PE fit because it affects employees' career development in various stages of the organization's life cycle [4]. In the pre-hire stage, the measurement of PE fit is often used for career counseling and job searching [5]. A majority of PE fit research has been conducted during the post-hire period, and studies show a strong linkage between PE fit and employees' attitude towards their jobs [6, 7].

Most PE fit studies have been conducted in the Western context and in industrial settings. In this study, we extend the PE fit theory to the health setting in China to examine the impact of PE fit among medical staff. Similar to the medical practice in Western countries, the medical practice in China often operates in teams, and thus relational coordination among medical professionals directly impacts on the quality of care [8, 9].

Among the four components of PE, PJ fit and PG fit are closely related concepts that concern the alignment of workers and their job as a team. PJ fit is defined as the consistency or match between the characteristics of a person and the work or tasks performed [10]. PJ fit contains two fundamental components, including demands-abilities fit and needs-supplies fit [1]. When an employee's knowledge, skills, and abilities (KSA) match the demand of work, the fit of demands-abilities occurs; when the work performed meets the employee's need, the fit of needs-supplies occurs [11]. Scroggins [12] added the concept of "self-concept of work fit" as an additional component to PJ fit. Unlike PJ fit that focuses on the consistency of individuals' capacities with tasks, PG fit concentrates on the compatibility between individuals and their workgroups [1]. PG fit exists when one shares similar or complementary values as those of team members or a person has a work-related KSA [3]. Compared with PJ-fit research, there is little research on antecedents of PG fit, and how it affects the performance of the team to which an individual belongs [13, 14].

Given the collaborative nature of the medical practice and the professionalism and particularity of the work itself [15], we focused our analysis on the potential effect of PJ and PG fit on job outcomes, particularly on job satisfaction and employees' intent to quit in Shanghai. Previous studies show that PJ and PG fit has a strong correlation with job satisfaction [1, 16]. However, this has not been analyzed and confirmed in China, and Shanghai provides a good setting for such an analysis. Shanghai is one of the cities with the most concentrated medical resources in China [17]. The reports showed that the average number of patients that a physician treats per day were 14.4 in Shanghai and hospital bed utilization were 95.85%. Medical professionals have a high workload [18].

In this study, we have not intended to assess the impact of PO fit. We had previously explored the impact of PO fit on job satisfaction and turnover intention among community health workers in China and found little impact of PO fit on the turnover intention [19]. It was also found that the difference in the PO fit of the survey subjects was small, which may be related to that they were all from public medical institutions [19]. What's more the values of individuals and organizations might be relatively difficult to change in a short time [20]. On the contrast, the PJ fit and PG fit are more modifiable to address attitude concerns among medical professionals, so as to improve the quality of medical care [21–24].

To help create an amicable environment for health professionals to practice and to improve quality of care, this paper aims to (1) examine the impact of PJ fit and PG fit on work attitudes of medical professionals in Shanghai; and (2) explore individual characteristics associated with PJ and PG fit.

To our knowledge, this is the first PJ fit and PG fit study among medical staff at the city level in China. It adds existing literature on the impact of PE fit on job outcomes among health professionals that require strong teamwork and under high pressure. Given a shortage and high turnover of health professionals in many developing countries, this study provides empirical evidence on the role of PE fit in improving job-related outcomes.

## Hypothesis

According to the input-process-output model (IPO), process experience (e.g., teamwork, coordination) is associated with individual and team outputs (e.g., job satisfaction, team performance, and professional efficacy) [25–27]: coordination and encouragement from team members enhance individual efficacy, which optimizes the collective group effort [25]. Previous studies also show that teamwork and job satisfaction are correlated [28–30]. Group settings can change individuals' motivation [31, 32], and group variables (e.g. group composition, leader and group characteristics) might affect individual members' sense of fulfillment towards their work [33]. Based on the theory of relationship coordination, it is also a form of organizing social capital, which can make it easier for people to obtain resources needed to accomplish one's work [34]. We know that having the resources needed to accomplish the work has been proved to be an important source of job satisfaction [35]. Medical practice is mostly team-based, requiring coordination of health providers within the same department or across different departments.

*Hypothesis 1*: Medical workers with higher PG fit have: (a) higher job satisfaction and (b) higher professional efficacy.

Empirical studies have shown that PJ fit has an important impact on employees' work attitudes, such as job satisfaction and turnover intention [36]. Smith [37] found that job satisfaction was largely determined by employees' work and their specific tasks. Caldwell [38] suggested that satisfaction and performance were enhanced when individuals select a job that was compatible with his or her traits and skills. These findings are consistent with the self-verification theory [39]. According to this theory, self-consistency improves the degree to which the individual feel that he can control and manipulate his surrounding environment. A stable self-concept allows individuals to negotiate social reality and understand how to act effectively in a given situation [39]. Also, the attraction–selection–attrition model showed that individuals will be attracted to and seek out jobs and organizations that provide them with meaningful self-confirming information and will likely continue in the job as long as self-confirming information is received and a high level of self-concept–job fit is perceived [40]. Meaningful work provides employees self-verified information, making the work itself attractive [41, 42]. PJ fit

has been established as an important predictor of job satisfaction and turnover intention [36], and we expect the same relationship will be held among medical staff in Shanghai.

*Hypothesis 2*: Medical workers with higher PJ fit have: (a) higher job satisfaction and (b) lower turnover intention.

Job satisfaction, reflecting the degree of personal enjoyment of work [43], has shown its relevance to both individual job outcomes and organizational results, and it is a stable predictor of employee turnover intention and behavior [44–47]. Built on these findings and in combination with Hypothesis 2 that PJ fit is associated with job satisfaction and job turnover directly, we further hypothesize that job satisfaction could serve as a mediator between PJ fit and turnover intention, and PJ fit has both direct or indirect association with the turnover intention. In fact, Hassan [48] and Chhabra [49] confirmed such relationships among employees in the Banking sector. Examining pathways on how PJ is associated with job turnover would help identify potential mitigation factors to address the concern of job turnover.

*Hypothesis 3*: Job satisfaction is a mediator for PJ fit and turnover intention, having direct or indirect association with medical staff's turnover intention.

## Methods

### Design

This study used data from the sixth National Health Service Survey (NHSS) conducted by the National Health Commission of China in 2018. One component of the survey was to investigate medical professionals' working environment and work attitudes in China. It measured PJ fit and PG fit, job satisfaction, turnover intention, and professional efficacy. The National Health Service Survey began in 1993 and has been conducted every five years since then.

We obtained the relevant NHSS data from Shanghai Municipal Health Information Center and used the survey data from Shanghai only for this study. The survey included all general hospitals and traditional Chinese medicine (TCM) general hospitals in 16 districts in Shanghai. All tertiary hospitals were included in the survey; 50% of secondary hospitals were randomly sampled and included in the survey; in each administrative district, 5 community health service centers were randomly selected. As to sampling health professionals, 10 clinicians and 5 nurses were randomly selected from each hospital, while 5 clinicians, 3 nurses, and 2 public health physicians were randomly selected from each community health service center. The survey contained 2,600 health professionals. A total of 2559 individuals were included in the analysis in this study after eliminating observations that had missing values and illogical errors from the sample.

### Participants

Table 1 shows the demographic information of the sample of 2559 participants. Women accounted for 72.76% of the sample, while participants aged under 40 years old represented 62.41%. In terms of the type of health facilities where health professionals worked, 1772 participants worked in hospitals (69.25%) and 787 worked in community health service centers (30.75%). The participants had different education levels. 86.83% of the medical workers had a bachelor's degree or above. The sample contained 1593 physicians, 871 nurses, and 95 other health personnel.

**Table 1. Demographic information for the study sample of Shanghai medical workers in Shanghai, 2018.**

| Item | Category | N | Percentage (%) |
|---|---|---|---|
| Sex | Male | 697 | 27.24 |
| | Female | 1862 | 72.76 |
| Age (years) | <30 | 444 | 17.35 |
| | 30–39 | 1153 | 45.06 |
| | 40–49 | 750 | 29.31 |
| | ≥50 | 212 | 8.28 |
| Marital status | Not married | 462 | 18.05 |
| | Married | 2097 | 81.95 |
| Education | Graduate degree | 738 | 28.84 |
| | Bachelor's degree | 1484 | 57.99 |
| | Junior college degree and below | 337 | 13.17 |
| Occupation | Physicians | 1593 | 62.25 |
| | Nurses | 871 | 34.04 |
| | Others | 95 | 3.71 |
| Type of organization | Hospitals | 1772 | 69.25 |
| | Community health service centers | 787 | 30.75 |

### Ethics approval and consent to participate

The current study was based on the secondary analysis of existing National Health Services Survey datasets. All data in this study were obtained and available for research upon the approval from authorized Shanghai Municipal Health Information Center. The data in this study did not contain the identifiable private information. The participants of original National Health Services Survey were adequately informed about all relevant aspects of the survey, including its objective and procedures and so on. The survey had obtained their informed consent.

### Measurements

Besides the demographic information, the NHSS also measured PJ and PG fit, job satisfaction, turnover intention, and professional efficacy. The detailed measurement for each of them is provided below.

### PJ fit

Measuring PJ fit is based on three aspects proposed by Scroggins in 2003, i.e., requirements–capability fit, self-concept–work fit, and supply–expectation fit, and revised for medical workers. The survey used an 8-item scale, each with a 7-point Likert scale (0 = complete nonconformity; 6 = completely consistent), to measure the PJ fit. One sample item is "My personality is a good match for this job." Cronbach's α coefficient of the scale was 0.921, suggesting a high reliability. The factor-loadings of the measured items using a factor analysis were 0.673–0.864, indicating a good construct validity of the scale.

### PG fit

The PG fit was measured using the scale constructed by Piasentin and Chapman (2007) and revised for medical workers. It included three aspects matching an individual and the team that the individual belonged to and was a 4-point Likert scale (1 = completely disagree; 4 = fully

agree). The scale had a reasonable reliability with a Cronbach's α coefficient of 0.768 and a good construct validity with the factor-loadings of the measured items being greater than 0.660.

## Job satisfaction, turnover intention, and professional efficacy

Job satisfaction was measured using the satisfaction scale developed by Brayfield and Rothe (1951) and revised for medical workers and the medical environment. It included five items: i.e., working environment, development prospects, welfare benefits, learning, and management status. The scale used a 6-point Likert scale (1 = completely disagree; 6 = fully agree). It also had a good reliability and construct validity. The Cronbach's α coefficient of the scale was 0.895, and the factor-loadings of the measured items were 0.664–0.875.

Turnover intention was measured through four items, which was revised based on Mobley (1978), and used a 6-point Likert scale (1 = completely disagree; 6 = fully agree). A sample item is "I often want to leave the industry I am working in currently." The Cronbach's α coefficient of the scale was 0.914.

Professional efficacy was based on the Maslach Burnout Inventory–General Survey (MBI-GS) scale but it was adapted to the nature of medical workers' work. A sample item is "My work will have a greater impact on the lives or happiness of others." The participants were asked to rate their agreement on a 4-point Likert scale (1 = completely disagree; 4 = fully agree). The Cronbach's α was 0.840. The factor-loadings of the measured items were higher than 0.65, indicating a reasonable construct validity of the scale.

## Analysis

The data was stored in Microsoft Access (Microsoft, Seattle, WA, USA), and we used the software of SPSS (Version 24.0; IBM Corp., Armonk, NY, USA) to perform descriptive statistical analyses and scale reliability tests, and calculate Cronbach's coefficients. Exploratory factor analysis was performed to examine the structural validity of the scales. AMOS software (Version. 24.0; IBM Corp.) was used for the confirmatory factor analysis and construct validity verification of the scale. Due to the lack of a consistent scale, we have carried out a unified transformation of these variables. Spearman correlation coefficients were estimated for observed variables, including PE fit, job satisfaction, turnover intention, and professional efficacy.

We also constructed a structural equation model to verify the research hypotheses and assess the impact of intermediating variables. Finally, a hierarchical regression model was used to examine the relationship between demographic variables (e.g., age, occupation, and education) and PJ and PG fit. We also checked the multicollinearity among independent variables and the equal variance of linear models used in this study. We found no issue of the multicollinearity among independent variables and heterogeneity of the models.

**Table 2. Average scores of PJ fit, PG fit, job satisfaction, turnover intention and professional efficacy.**

| Items | Means | SD |
|---|---|---|
| PJ fit | 3.54 | .78 |
| PG fit | 3.89 | .67 |
| Job satisfaction | 3.29 | .86 |
| Turnover intention | 2.30 | 1.03 |
| Professional efficacy | 4.30 | .69 |

# Results

## Descriptive statistics

The descriptive statistical analysis shows that the average scores of PJ fit and PG fit among the survey subjects were 3.54±0.78 and 3.89±0.67, respectively. The scores of job satisfaction, turnover intention, and professional efficacy were 3.29±0.86, 2.30±1.03 and 4.30±0.69 respectively (Table 2).

## Hypothesis testing

Table 3 shows the correlation among the five main research indicators. Except for turnover intention and professional efficacy, all correlation coefficients were statistically significant ($P < 0.001$). PJ fit ($r = -0.437$), PG fit ($r = -0.347$), and job satisfaction ($r = -0.468$) were negatively correlated with turnover intention. PJ fit ($r = 0.561$), PG fit ($r = 0.434$), and professional efficacy ($r = 0.129$) were positively correlated with job satisfaction. PJ fit ($r = 0.310$) and PG fit ($r = 0.385$) were positively correlated with professional efficacy.

Fig 1 shows the relationship among the five variables using a structural equation model. The model considered the direct impact of PG fit on job satisfaction and professional efficacy, the direct impact of PJ fit on job satisfaction and turnover intention, and the mediating effect between PJ fit and turnover intention by job satisfaction (Fig 1). The measures of goodness of fit suggest the model fitted the data quite well (the chi-squared statistic divided by the degrees of freedom = 6.438; the root mean square error of approximation = 0.046; goodness-of-fit index = 0.950; adjusted goodness-of-fit index = 0.936; Tucker-Lewis index = 0.962).

Table 4 shows that PG fit was directly associated with job satisfaction and professional efficacy ($\beta = 0.338$, $P < 0.001$ and $\beta = 0.518$, $P < 0.001$, respectively). The result supported Hypothesis 1. Additionally, PJ fit was directly associated with job satisfaction ($\beta = 0.470$, $P < 0.001$) and turnover intention ($\beta = -0.206$, $P < 0.001$), and the result supported Hypothesis 2. To validate Hypothesis 3, we constructed a pathway using job satisfaction as a mediating factor between PJ fit and turnover intention. We found that PJ fit was indirectly associated with job turnover intention, and the association was statistically significant ($\beta = -0.185$, $P < 0.001$), which supported Hypothesis 3.

In the hierarchical regression analysis to explore factors associated with PJ fit and PG fit, the PJ fit model (Model 2) included two variables: age and occupation, with $R^2 = 0.010$, $F = 12.551$ ($P < 0.001$), and $\Delta R^2 = 0.004$. After including the variable of education, $R^2$ had not been changed, and the P-value of education became 0.179. Thus, education was excluded from the PJ fit analysis. The PG fit model (Model 5) included occupation, education, and age: $R^2 = 0.032$, $F = 28.247$ ($P < 0.001$), and $\Delta R^2 = 0.005$ (Table 5).

The results showed that age was positively associated with PJ fit but negatively associated with PG fit. Education also had a negative association with PG fit. Specifically, younger

**Table 3. Correlations of indicators for Shanghai medical workers in 2018.**

| Correlations | PJ fit | PG fit | Job satisfaction | Turnover intention | Professional efficacy |
|---|---|---|---|---|---|
| PJ fit | 1.00 | | | | |
| PG fit | 0.382*** | 1.00 | | | |
| Job satisfaction | 0.561*** | 0.434*** | 1.00 | | |
| Turnover intention | −0.437*** | −0.347*** | −0.468*** | 1.00 | |
| Professional efficacy | 0.310*** | 0.385*** | 0.129*** | −0.086 | 1.00 |

*** $p < 0.001$ (2-tailed).

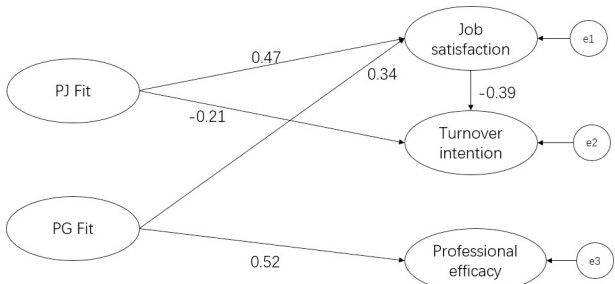

**Fig 1. Structural equation model diagram produced using AMOS software.** Observed variables are omitted from the graph. The model contains latent variables and e1, e2, and e3 are error items. The arrow indicates the regression path from the independent variables to the dependent variables.

medical workers generally had a lower level of PJ fit but had a higher level of PG fit. And those who had a higher level of education tended to have a lower level of PG fit. Among all occupations, physicians had a lower PJ fit and PG fit than other medical professionals.

## Discussion

In keeping with the PE fit theory, we introduce a theoretical model that present hypotheses about the relations of PJ and PG fits with job satisfaction, turnover intention, and professional efficacy. Additionally, we explored the mediation mechanism of job satisfaction in PJ fit on turnover intention.

This study found that the PJ fit of medical workers showed a strong correlation with job satisfaction and turnover intention. This result is almost identical to a study conducted in US practitioners by Hinami et al., which showed a close association between PJ fit and job satisfaction (r = 0.570, P<0.001) [42]. The result is also consistent with the finding from a meta-analysis that shows the correlation coefficient between PJ fit and job satisfaction of 0.56 and the correlation coefficient between PJ fit and turnover intention of −0.46 [1]. As to job satisfaction, the meta-analysis found a correlation coefficient of 0.31 for PG fit. Our corresponding estimate is higher. Although some studies found that PJ fit has a direct effect on job satisfaction, it does not have a significant effect on turnover intention [50]. Overall, most studies have confirmed the relationships between PJ fit, turnover intention, and turnover behavior [51–53], and our research findings are consistent with the relationships identified from the literature.

We also found that PG fit was directly associated with professional efficacy. A study found that there was a strong correlation between PO fit and professional efficacy in nonmedical fields [54]. Few studies so far have examined the impact of PE fit on professional efficacy. With

**Table 4. Path decomposition of effects in structural equation modeling.**

|  | Total effect | Direct effect | Indirect effect |
|---|---|---|---|
| Job satisfaction ← PJ fit | 0.470*** | 0.470*** |  |
| Job satisfaction ← PG fit | 0.338*** | 0.338*** |  |
| Professional efficacy ← PG fit | 0.518*** | 0.518*** |  |
| Turnover intention ← PJ fit | −0.391*** | −0.206*** | −0.185*** |
| Turnover intention ← PG fit |  |  | −0.133*** |
| Turnover intention ← Job satisfaction | −0.393*** | −0.393*** |  |

***Relationship pathway was statistically significant (P<0.001).

**Table 5. Hierarchical regression of PJ and PG fits.**

| Variables | PJ fit | | | | PG fit | | | | | |
|---|---|---|---|---|---|---|---|---|---|---|
| | Model 1 | | Model 2 | | Model 3 | | Model 4 | | Model 5 | |
| | Estimate | Standardized estimate | Estimate | Standardized estimate | Estimate | Standardized estimate | Estimate | Standardized estimate | Estimate | Standardized estimate |
| Intercept | 3.256*** | | 3.107*** | | 3.644*** | | 3.486*** | | 3.618*** | |
| Age | 0.008*** | 0.079 | 0.009*** | 0.088 | | | | | −0.054*** | −0.069 |
| Occupation | | | 0.083** | 0.060 | 0.17*** | .143 | 0.126*** | 0.106 | 0.113*** | 0.095 |
| Education | | | | | | | 0.079*** | 0.091 | 0.083*** | 0.096 |
| $R^2$ | 0.006 | | 0.010 | | 0.021 | | 0.027 | | 0.032 | |
| F | 15.977*** | | 12.551*** | | 53.582*** | | 36.097*** | | 28.247*** | |
| $\Delta R^2$ | 0.006 | | 0.004 | | 0.021 | | 0.007 | | 0.005 | |
| $\Delta F$ | 15.977*** | | 9.076** | | 53.582*** | | 18.252*** | | 12.230*** | |

***$P < 0.001$;

**$P < 0.005$.

the introduction of the concept of team-based care, in-depth studies of PG fit are particularly urgent.

Additionally, the results on the hierarchical regression of PJ and PG fits show that younger medical professionals generally have a lower level of PJ fit and physicians tend to have a lower level of PG fit than other medical professionals. Furthermore, a lower level of PJ fit was more pronounced among younger physicians, suggesting that young physicians, especially those who have just started their work, are the main group who need to be targeted for improving the PJ fit. This group of young physicians often just go through the transition from medical students into medical clinicians. Medical students have a longer training period than other graduate students. This higher training cost and long training period lead to higher expectations of being medical clinicians by medical students, resulting in a gap between their expectations and the reality [55]. Additionally, new employees tend to be less adapted to a complex clinical environment and lack the experience in dealing with complicated physician-patient relationships in China [56].

We also found that older physicians with higher education had a lower level of PG fit. The possible reason for the lower PG fit of highly qualified physicians is that with the increase in age among senior medical professionals, their promotion opportunities are decreasing. This would result in a lack of incentives for them to continue their work. Nevertheless, these physicians have a long working life and have a set of habitual work patterns and work experience. Thus, they may find it difficult to get out of their comfort zone and to ensure the consistency with the values of their teams.

## Limitations

This study had three main limitations. First, the number of attitude variables included in this study was limited and it was difficult to provide a comprehensive picture of job outcomes among medical professionals using PE theory. Second, considering Shanghai's socio-economic status and the extremely high workload of health professionals, the results from this study may not be generalizable to medical professionals in other places in China. It is hoped that a larger range of data can be collected for multicenter comparison in future studies. Third, we used self-reported questionnaires (tools) to measure key research variables in this study, and the reported results may be subjective to potential bias. For example, although we have tested the

construct validity of questionnaires, participants can still provide untruthful responses, and it is not clear if the questioners used in this study have a good criteria validity to capture the essence of what we would like to measure. Future research should validate the questionnaires from other perspectives (e.g. criteria validity and content validity).

## Implications

This was the first PJ fit and PG fit study among health professionals at the city level in China. The results of this study show that PJ fit is a critical factor determining job satisfaction and job turnover. It is important for hospital administrators to identify where the lack of fit lies between medical workers and work and to address it (e.g. through training), thereby reducing employee dissatisfaction and turnover intention. The turnover intention of medical staff was at a high level in China [57]. Ensuring PE fit of health professionals might alleviate the persistent problems of labor shortage and frequent turnover of medical staff in China. In addition, the work attitude of medical staff is important in determining the quality of medical care. Identifying medical staff who fit the job and share similar values with peers may improve the quality of. Care and ease the strained doctor-patient relationship in China.

In addition to the measurement of PJ fit when recruiting medical workers, it is also important to focus on strengthening the relevant fits among young physicians. It is possible to choose a suitable position after rotation for individual employees by assessing the degree of fit between their job interests and the nature of their work. The medical professional association within hospitals or health facilities can provide a communication platform such as social media for junior and senior physicians, where senior physicians can share tacit knowledge according to the physicians' perspectives and experiences [58] and self-improvement methods with junior physicians through pairing or group sharing to improve their PJ fit.

Although the turnover intention may not be consistent with the actual behavior, it is a prerequisite for the behavior. Turnover research has shown that an employee's self-expressed intentions to leave their job are the best predictor of actual turnover [59]. Given prevalent issues of a shortage and high turnover of health professionals in many developing countries and the strong association between PJ fit and job turnover intention, it is important to deepen research on PJ fit in order to better address these chronic problems.

We also found that the increase of PG fit can improve professional efficacy and job satisfaction, but older physicians with higher academic qualifications tend to have lower PG fit. It is recommended that team-building training be conducted in a targeted manner. Health administrators can, for example, establish workgroups that allow them to take advantage of their being in a diverse workforce to foster collaboration. While health professionals who have better education with long working experience could bring a wealth of experience to the team, health administrators should also encourage them to absorb new ideas or values to achieve a higher degree of fit to the team.

## Conclusions

The results show that medical workers with higher PJ or PG fits have higher job satisfaction and those with higher PG fit have higher professional efficacy. At the same time, the impact of PJ fit on medical workers' turnover intention was influenced to a certain extent by job satisfaction. And we also found that poor PJ fit was more pronounced among younger physicians, while older physicians with higher education have poor PG fit. Health administrators should consider changing management mechanisms to effectively promote medical professionals' PJ and PG fit.

## Supporting information

**S1 Dataset.**
(XLSX)

**S2 Dataset.**
(XLSX)

**S1 Questionnaire.**
(DOCX)

## Acknowledgments

We are very grateful to the reviewers and editors for their valuable comments and corrections, which made our manuscript more optimized. At the same time, we also thank to Centre of Health Statistics and Information, Shanghai Municipal of Health Commission for providing research data.

## Author Contributions

**Conceptualization:** Yuyin Xiao, Chenshu Shi, Guohong Li.

**Data curation:** Yuyin Xiao, Minye Dong.

**Formal analysis:** Yuyin Xiao, Chenshu Shi.

**Investigation:** Zhenyi Shao, Hua Xie.

**Methodology:** Yuyin Xiao, Minye Dong, Chenshu Shi, Guohong Li.

**Project administration:** Zhenyi Shao.

**Resources:** Hua Xie.

**Software:** Minye Dong.

**Writing – original draft:** Yuyin Xiao, Minye Dong, Wu Zeng.

**Writing – review & editing:** Chenshu Shi, Wu Zeng, Zhenyi Shao, Hua Xie, Guohong Li.

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
