## [Decision Letter · Decision Letter 0]

11 Jun 2020

PONE-D-20-01262

How does person–environment fit affect Shanghai medical workers’ job satisfaction, turnover intention, and sense of accomplishment? An exploratory cross-sectional study

PLOS ONE

Dear Dr. Li,

Thank you for submitting your manuscript to PLOS ONE. After careful consideration, we feel that it has merit but does not fully meet PLOS ONE’s publication criteria as it currently stands. Therefore, we invite you to submit a revised version of the manuscript that addresses the points raised during the review process.

We look forward to receiving your revised manuscript.

Kind regards,

Geilson Lima Santana, M.D., Ph.D.

Academic Editor

PLOS ONE

Journal Requirements:

3. Thank you for incuding your ethics statement on the submission form: "N/A". We note in the Methods section of your manuscript you state that: "Ethics approval and consent to participate Ethics Review Board of Shanghai Centre of Health Statistics and Information, Ministry of Health. All respondents provided

written or verbal informed consent before being interviewed."

a) Please amend your current ethics statement to confirm that your named institutional review board or ethics committee specifically approved this study.

5. Your ethics statement must appear in the Methods section of your manuscript. If your ethics statement is written in any section besides the Methods, please move it to the Methods section and delete it from any other section. Please also ensure that your ethics statement is included in your manuscript, as the ethics section of your online submission will not be published alongside your manuscript.

Reviewers' comments:

Reviewer's Responses to Questions

**Comments to the Author**

1. Is the manuscript technically sound, and do the data support the conclusions?

Reviewer #1: Partly

Reviewer #2: Partly

2. Has the statistical analysis been performed appropriately and rigorously? 

Reviewer #1: I Don't Know

Reviewer #2: No

3. Have the authors made all data underlying the findings in their manuscript fully available?

Reviewer #1: Yes

Reviewer #2: Yes

4. Is the manuscript presented in an intelligible fashion and written in standard English?

Reviewer #1: Yes

Reviewer #2: Yes

5. Review Comments to the Author

Reviewer #1: General comments:

The study explored the attitudes of Shanghai medical workers based on the PE fit theory and explore the intrinsic relationships and influential factors between attitudes, turnover intention, and sense of accomplishment.This study is really intersting, but what is your innovations and new discoveries? Or is it just to prove that this model is correct? Maybe it needs more clear to express this part.

So many terms makes readers difficult to understand the logic and correlation between different terms. It needs to reorganise to make it more clear. Such job satisfaction, the measurement of job satisfaction included the environments, so what is difference between other relvant terms of envirnments?

Detail comments:

Line 107-110 "However, it is worth noting that if people’s respect and self-worth needs are not met, it will lead to a decline in their sense of accomplishment. This will not cause dissatisfaction and there will be no sense of burnout". This sentence seems like the part of dissussion. The logic of literature review is not clear and it needs to reorganise.

Line 123-124 Theories of PE interaction were very influential in management studies. Maybe it needs to delete.

Line 123-136 This part is too tedious. It should be more concise.

Line 161 randomly selected according to the proportion of 5 for each district. Please recheck the grammer and number.

Line 162 first-line medical workers, what is meaning of first-line medical workers, how to distinguish other types of medical workers?

Line 161-165 How did you guarantee randomness?

Line 165-166 What is the number of person who refused the survey?

Table 1 The definition of Physicians in China is very different from that in this study. Please recheck the definition.

Discussion part

Line 35-37 How did you make such findings? It did not appear in your tables.

Line 41-43 " At the same time, new employees tend to be less adapted to the complex clinical environment and lack experience in dealing with physician–patient relationships and coordinating resources. " Can the SEM make such conclusions?

The descriptions of PE fit is not clear, it should be desribed usuing more detail words, not professional terms.

Reviewer #2: In this manuscript, the authors aim to explore the relations between two components of person-environment fit theory (namely Person-group fit and person-job fit) and employee wellbeing outcomes of job satisfaction, turnover intention and sense of accomplishment. The final aim of the authors is to provide recommendations for improvement and intervention directed to both health care employees and employers. To achieve this, the authors analyzed cross-sectional data from a large survey delivered to health care employees in Shanghai, using correlation analysis, and other bivariate techniques such as hierarchical regressions to test the hypotheses. As a strength of the manuscript, the authors present analyses from a large data set and propose the relation between some variables that could lead to practical implications as is their aim. However, I do not recommend the publication of this manuscript until the authors address several major aspects, which affect the internal and external validity of their study and therefore the validity of their conclusions. The authors should also consider other minor revisions.

Regarding the major aspects that authors should review: First, though the relevance of the problem they are addressing seems obvious, the authors are not being explicit about it, nor creating a clear link between the problem they want to address and the solution they aim to provide. In this sense, the authors claim they propose to bring empirical evidence by exploring “the attitudes of Shanghai medical workers based on the PE fit theory and explore the intrinsic relationships and influential factors between attitudes, such as job satisfaction, turnover intention, and sense of accomplishment” (page 5, second paragraph). However when justifying the relevance of the study they claim the need for theoretical development, which they do not provide in the manuscript (see the end of page 3: “it is of theoretical significance to further explore the antecedents of job satisfaction of medical workers”)

Second, the authors should provide a deeper and more accurate revision of the theoretical background of their study. Specifically they need to review the concepts of Personal Accomplishment, turnover and Person-Environment fit. Regarding Personal Accomplishment, I suggest the authors review in depth the theory of burnout proposed by Maslach, which they use to justify the construct of personal accomplishment. In this sense, the authors writing leads to misinterpretations on the relation between different constructs. As an example of this, in page 10 third paragraph the authors report: “Professional accomplishment and burnout are opposite definitions, but they have some relationship”. However, according to extant literature on burnout, personal accomplishment is one of burnout dimensions, because of this it is clear they are related. Thus, I suggest authors review again Maslach et al 2001 and Maslach and Leiter, 2008, which are part of the references. The review the authors make on these components must include also an examination on the instrument they are using to assess personal accomplishment, which comes from the Maslach Burnout Inventory-General Survey. As the authors should report in their manuscript, this version of the MBI does not assess Personal Accomplishment, but a different construct named “Professional Efficacy” which is less related to a sense of personal fulfilment and more related to a job related sense of self-efficacy. In this sense, their results are not related to accomplishment as they claim, but to professional efficacy. A second example of some of the confusions the writing of the manuscript leads to is that, to my knowledge, no literature propose any relation between Maslow’s theory of motivation and burnout dimension of professional efficacy (or personal accomplishment). If the authors do know any literature that propose these links I urge them to make proper cites and use the references, otherwise they should further develop their reasoning to propose such links. Regarding the literature on turnover and Person-Environment fit, the authors should devote a portion of the introduction and the theoretical background to define the constructs and include relevant empirical research that is pertinent to the development of their hypotheses.

Third, the authors will want to review the development of their hypotheses and provide further justification for each one of them. For example, it is not clear why in H1 the authors look for a relationship between Person-Group fit and the dependent variables of job satisfaction and accomplishment, whereas in H2 they look for a relationship between Person-Job fit and the outcomes of job satisfaction and turnover intention. Thus, it is not clear why H1 does not include turnover intention in their predictions and why H2 does not include accomplishment. Neither has it been clear why the authors proposed a mediation hypothesis in H3. The hypotheses should be driven from previous theory and empirical research that the authors articulate with their own aim. One way of achieving this is by addressing the second major revision, which was presented in the previous paragraph.

Fourth, the pertinence of the analyses performed, and therefore the results presented is not clear. Also because of the previously named issues related to major problems in the hypotheses, it cannot be said that the results support the hypotheses (this is because it is not clear what the authors are hypothesizing about). Some recommendations to improve this aspect include: (i) The authors should provide specifications on the type of factor analyses performed. (ii) There is no presentation of dependent and independent variables descriptives (The descriptive statistics presented are the description of the sample, this should be moved to the participants subsection in the methods section). (iii) The authors should consider better analyses approaches, such as multivariate and regressions that include control variables such as age, gender and tenure (this is only presented in the hierarchical regression models, but the results are not clearly explained. (iv)Hierarchical regression models for PJ fit should also include education level, or specify why it does not consider it

Some of the minor revisions are:

1. Review the need to make an introduction explaining attitudes in psychology, at the moment it seems unnecessary.

2. The authors should make explicit how prejudice was minimized (as proposed in page 7).

3. In page 7, paragraph 2 “physicians were randomly selected from the community health service center according to the number of workers”. It should be explained in greater depth how this selection was done and how many workers were part of these centers.

4. No explanation on how the participants were recruited is provided.

5. Example of the items that were develop for PJ and PG constructs should be provided.

6. Regarding the analyses performed major clarity and justification of the analyses performed should be presented (specially regarding the structural equation model).

7. Descriptive statistics for the dependent and independent variables must the presented.

8. The discussion should not include results.

9. The implications of the study should be further explored.

10. The limitations of the study should include the limitations of self-reported questionnaires.

6. PLOS authors have the option to publish the peer review history of their article (what does this mean?). If published, this will include your full peer review and any attached files.

Reviewer #1: No

Reviewer #2: Yes: Camila Umaña

---

## [Author Response · Author response to Decision Letter 0]

23 Aug 2020

Dear editor Geilson Lima Santana,

We are glad to receive your letter for the revision request. Manuscript ID PONE-D-20-01262 entitled "How does person–environment fit affect Shanghai medical workers’ job satisfaction, turnover intention, and sense of accomplishment? An exploratory cross-sectional study" which we submitted to PLOS ONE, has been revised. Thank you for your hard work and the comments from two reviewers. According to your request and reviewers’ advice, we have made a careful modification of our manuscript. The following is the list and explanation of the revision.

The submitted manuscript has been modified according to PLOS ONE's style requirements.

According to your request, we have attached the questionnaire in the supplementary information. However, it should be noted that this is the Shanghai part of the sixth national health service survey and the questionnaire are not allowed to be disclosed. For the needs of article review, we just provide the questionnaire to the editors and reviewers. Please keep it only for this manuscript, thanks.

We have modified the ethics section in the online submission.

According to your request, we have submitted all the data involved in this study in the supplementary information.

We have amended the ethics statements in the Methods section of the manuscript. (see the Ethics approval and consent to participate part in red）

Response to the reviewer 1：

Thank you very much for your advice. According to your advice, we have made a careful modification of our manuscript. And all the revised parts are marked in red in our paper. The following is the explanation of the revision.

1. The study explored the attitudes of Shanghai medical workers based on the PE fit theory and explore the intrinsic relationships and influential factors between attitudes, turnover intention, and sense of accomplishment. This study is really intersting, but what is your innovations and new discoveries? Or is it just to prove that this model is correct? Maybe it needs more clear to express this part.

We hope to explore the antecedents that affect the attitude variables of medical staff, and provide managers with implementation paths that can be improved and intervened. According to the research reviews, it is found that the theory of person-environment fit is often used to explore the impact on attitude variables. Based on the previous research on P-O fit, this study hopes to further understand the impact of P-J fit and P-G fit on attitude variables. There are few cross-sectional studies on applying PE fit theory to Chinese medical staff. The results of this study also have practical significance for strengthening the management of clinical medical staff. (See the Background and Hypothesis parts in red)

2. So many terms makes readers difficult to understand the logic and correlation between different terms. It needs to reorganise to make it more clear. Such job satisfaction, the measurement of job satisfaction included the environments, so what is difference between other relvant terms of envirnments?

Thank you for your suggestion. It is true that the logical relationship in the previous version of the manuscript is not clear enough. For this reason, we have rewritten the content of the Background and Hypothesis section, hoping that it will be easy for readers to understand. The question concerning the environment in the measurement of satisfaction is "How satisfied are you with the working environment?" The working environment here is a relatively general concept. In the theory of person-environment fit, the specific PJ fit and PG fit scales are used to evaluate the fit between person with their job, and person with their group. They are two important components in the theory of fit between person and environment. (See the Background and Hypothesis parts in red)

3. Line 107-110 "However, it is worth noting that if people’s respect and self-worth needs are not met, it will lead to a decline in their sense of accomplishment. This will not cause dissatisfaction and there will be no sense of burnout". This sentence seems like the part of discussion. The logic of literature review is not clear and it needs to reorganise.

Based on your suggestions, we sorted out the logical relationships and rewrote the content of this part. (See the Hypothesis parts in red)

4. Line 123-124 Theories of PE interaction were very influential in management studies. Maybe it needs to delete.

Thanks for your suggestions, we have removed.

5. Line 123-136 This part is too tedious. It should be more concise. 

We have rewritten the content of this part. (See the Hypothesis parts in red)

6. Line 161 randomly selected according to the proportion of 5 for each district. Please recheck the grammer and number.

According to your advice, we have revised the sentence. (See the paragraph 2 of Design in red)

7. Line 162 first-line medical workers, what is meaning of first-line medical workers, how to distinguish other types of medical workers?

First-line medical workers or frontline medical workers refer to clinicians and nurses who are in close contact with patients in clinical work, and are different from those engaged in scientific research and managers in hospitals. This may be due to the problem of non-native language expressions. In order to facilitate understanding, we have modified into clinical medical workers.

8. Line 161-165 How did you guarantee randomness?

On the one hand, it is a large-scale survey at the national level, the randomness of sampling is completely guaranteed. On the other hand, we use Runs test to verify the randomness of the sample. In accordance with the principle of stratified sampling, we grouped administrative regions and types of institutions, and introduced age as the time variable. We established research hypotheses separately to verify the randomization of each group. It was found that the P value of each group was greater than 0.05. 

9. Line 165-166 What is the number of person who refused the survey? 

We hope to answer your question. However, it is a national-level investigation, and we are not authorized to disclose this information. Under normal circumstances, the investigation conducted by the National Health Commission will be highly cooperative.

10. Table 1 The definition of Physicians in China is very different from that in this study. Please recheck the definition.

The classification in Table 1 is more accurately described as qualification. In order to unify with the existing research, we merged it and modified the classification method involved in the subsequent analysis. (See the Table1 in red)

11. Line 35-37 How did you make such findings? It did not appear in your tables. 

Through hierarchical regression analysis, we included the age variable in model, and according to the results we found age had a positive predictive effect on PJ fit and a negative predictive effect on PG fit. In order to fully explain the results of the study, we have supplemented the interpretation of the results. (See the last paragraph of Results in red) 

12. Line 41-43 " At the same time, new employees tend to be less adapted to the complex clinical environment and lack experience in dealing with physician–patient relationships and coordinating resources. " Can the SEM make such conclusions?

According to the results of stratified regression, we found that younger physicians had lower PJ-fit. The above sentence is speculating on the possible reasons for this result, and the purpose is to provide a basis for subsequent intervention and improvement measures.

13. The descriptions of PE fit is not clear, it should be desribed usuing more detail words, not professional terms.

We try to clarify the concept of PE fit in the revised manuscript and the different dimensions it contains as much as possible. Hoping it can help readers better understand the meaning.

Response to the reviewer 2:

Thank you very much for your detailed instructions and suggestions. According to your advice, we have made a careful modification of our manuscript. And all the revised parts are marked in red in our paper. The following is the explanation of the revision.

1. First, though the relevance of the problem they are addressing seems obvious, the authors are not being explicit about it, nor creating a clear link between the problem they want to address and the solution they aim to provide. In this sense, the authors claim they propose to bring empirical evidence by exploring “the attitudes of Shanghai medical workers based on the PE fit theory and explore the intrinsic relationships and influential factors between attitudes, such as job satisfaction, turnover intention, and sense of accomplishment” (page 5, second paragraph). However when justifying the relevance of the study they claim the need for theoretical development, which they do not provide in the manuscript (see the end of page 3: “it is of theoretical significance to further explore the antecedents of job satisfaction of medical workers”)

According to your suggestion, in order to further clarify the problem to be solved and the internal relationship between variables, we have reorganized the background and the content of the research hypothesis, and supplemented the source and basis of the research hypothesis one by one. We hope that readers can better understand the purpose and content of the research. 

First of all, we put forward the concept of PE fit and its prediction of the variable of working attitude. Then we clarify the focus of this study on PJ fit and PG fit; we explain the concept and development of PJ fit and PG fit respectively. In the research hypothesis section, we explained the basis and development of each hypothesis and the specific content in our research. (See the Background and Hypothesis parts in red)

2. Second, the authors should provide a deeper and more accurate revision of the theoretical background of their study. Specifically they need to review the concepts of Personal Accomplishment, turnover and Person-Environment fit. Regarding Personal Accomplishment, I suggest the authors review in depth the theory of burnout proposed by Maslach, which they use to justify the construct of personal accomplishment. In this sense, the authors writing leads to misinterpretations on the relation between different constructs. As an example of this, in page 10 third paragraph the authors report: “Professional accomplishment and burnout are opposite definitions, but they have some relationship”. However, according to extant literature on burnout, personal accomplishment is one of burnout dimensions, because of this it is clear they are related. Thus, I suggest authors review again Maslach et al 2001 and Maslach and Leiter, 2008, which are part of the references. The review the authors make on these components must include also an examination on the instrument they are using to assess personal accomplishment, which comes from the Maslach Burnout Inventory-General Survey. As the authors should report in their manuscript, this version of the MBI does not assess Personal Accomplishment, but a different construct named “Professional Efficacy” which is less related to a sense of personal fulfilment and more related to a job related sense of self-efficacy. In this sense, their results are not related to accomplishment as they claim, but to professional efficacy. A second example of some of the confusions the writing of the manuscript leads to is that, to my knowledge, no literature propose any relation between Maslow’s theory of motivation and burnout dimension of professional efficacy (or personal accomplishment). If the authors do know any literature that propose these links I urge them to make proper cites and use the references, otherwise they should further develop their reasoning to propose such links. Regarding the literature on turnover and Person-Environment fit, the authors should devote a portion of the introduction and the theoretical background to define the constructs and include relevant empirical research that is pertinent to the development of their hypotheses.

Probably due to non-native writing, the expression in the previous manuscript may indeed be problematic. We apologize if it is difficult for you or other readers to understand or cause misunderstanding. Thank you very much for your detailed and professional advice. In the revised manuscript, we have improved the concept of variables and the description of the relationship between variables, and at the same time supplemented the relevant research content of the research hypothesis. Regarding the part of job burnout, we must admit that this is indeed one of the limitations of this study. In our questionnaire, only one dimension of job burnout is involved (that is sense of accomplishment). Therefore, this study only discusses the impact of PE theory on a single dimension of job burnout. Since it is a national survey, we failed to add two other dimensions to the questionnaires. However, we hope to improve it in our future research. Regarding the relationship between Maslow’s theory of motivation and burnout as you mentioned, we have indeed found some related studies. ([1] Hale A J, Ricotta D N, Freed J, et al. Adapting Maslow's hierarchy of needs as a framework for resident wellness. Teaching and learning in medicine, 2019, 31(1): 109-118.[2] Gao Hong. Looking at nurses’ job burnout from Maslow’s hierarchy of needs theory[J]. Contemporary Medical Essays (Chinese), 2015, 13(16): 3-4.) Due to the re-writing of the background part, we have adjusted this content appropriately.

3. Third, the authors will want to review the development of their hypotheses and provide further justification for each one of them. For example, it is not clear why in H1 the authors look for a relationship between Person-Group fit and the dependent variables of job satisfaction and accomplishment, whereas in H2 they look for a relationship between Person-Job fit and the outcomes of job satisfaction and turnover intention. Thus, it is not clear why H1 does not include turnover intention in their predictions and why H2 does not include accomplishment. Neither has it been clear why the authors proposed a mediation hypothesis in H3. The hypotheses should be driven from previous theory and empirical research that the authors articulate with their own aim. One way of achieving this is by addressing the second major revision, which was presented in the previous paragraph.

Thank you for your valuable suggestions. We have rewritten and sorted out the content of the research hypothesis, hoping to explain as much as possible why we determined the above research hypothesis. (See the Hypothesis parts in red)

4. Fourth, the pertinence of the analyses performed, and therefore the results presented is not clear. Also because of the previously named issues related to major problems in the hypotheses, it cannot be said that the results support the hypotheses (this is because it is not clear what the authors are hypothesizing about). Some recommendations to improve this aspect include: (i) The authors should provide specifications on the type of factor analyses performed. (ii) There is no presentation of dependent and independent variables descriptives (The descriptive statistics presented are the description of the sample, this should be moved to the participants subsection in the methods section). (iii) The authors should consider better analyses approaches, such as multivariate and regressions that include control variables such as age, gender and tenure (this is only presented in the hierarchical regression models, but the results are not clearly explained. (iv)Hierarchical regression models for PJ fit should also include education level, or specify why it does not consider it.

(i) The results of the reliability and validity analysis of each dimension have been presented in the Instruments section.

(ii) We added a statistical description of the dependent and independent variables in the results section and moved the sample description to the methodology section. (see the paragraph 1 of Results and Participants of Methods in red)

(iii) The reason why it was not included as control variables is that after a large number of literature reviews, we did not find that these variables have an absolute impact on PG fit, PJ fit, job satisfaction, turnover intention, and sense of accomplishment. Nevertheless, we still use generalized sensitivity analysis to explore the impact of omit variables. According to each research hypothesis, we test the effect values of the omit variables separately. (Fig 1-3) Because the value of standardized coefficients of unobserved confounders in Hypothesis 1(b) and Hypothesis 2(a) are too small, it is unable to generate related graphs of GSA. The results of the sensitivity analysis show that the variables of age, gender, education, and occupation have very small impact. Finally, we incorporate these four variables into the structural equation model as control variables. The results of model fitting before and after adding control variables have been shown in Table 1. We found that the results after including the control variables were not better than before. We speculate that this may be related to the study design. Since the study adopted the principle of stratified random sampling and strictly adopted randomization in the sample selection process, the influence of covariates was not significant in this study. We simplifed the above process and add it to the methodology section. (See the Analysis in red)

Figure 1 GSA of Hypothesis I (a)

Figure 2 GSA of Hypothesis 2 (b)

Figure 3 GSA of Hypothesis 3

Table 1 The results of model fitting before and after adding control variables

 Before After

CMIN/DF 6.438 8.063

RMSEA 0.046 0.053

GFI 0.950 0.929

AGFI 0.936 0.912

TLI 0.962 0.935

CFI 0.965 0.943

(iv) In the analysis process, we considered the three variables of age, occupations and education. But after adding the variable of education, the value of R2 had not been changed, and the significance of this variable was 0.179. Therefore, the final model does not include the variable of education. We have supplemented the above content in the results section, which will help the reader understand.

5. Review the need to make an introduction explaining attitudes in psychology, at the moment it seems unnecessary.

Due to the rewriting of the background section, this content has been deleted according to your suggestions.

6. The authors should make explicit how prejudice was minimized (as proposed in page 7).

The data for the analysis were drawn from the Sixth National Health Service Survey. The National Health Service Survey began in 1993 and is conducted every five years. The survey is organized by the National Health Commission, the Planning Department is responsible for coordination, and the National Statistical Information Center is responsible for the organization and implementation. They organize investigators to conduct unified training, investigation, data entry, quality control, supervision, etc. In order to fully grasp the situation in China, several rounds of expert consultations have been conducted to revise the survey plan to minimize prejudice and strictly implement the principle of randomized sampling. The investigators were systematically trained before the survey process. No hospital supervisor was involved in the whole process to ensure the authenticity of the questionnaire data.

7. In page 7, paragraph 2 “physicians were randomly selected from the community health service center according to the number of workers”. It should be explained in greater depth how this selection was done and how many workers were part of these centers.

The data for the analysis were drawn from the Sixth National Health Service Survey, which was provided by Shanghai Municipal Health Information Center. Because we were not involved in the sampling process, there is no more detailed information to provide, and also we were not informed about the number of employees in each center. However it is a large-scale survey at the national level, the randomness of sampling has been completely guaranteed.

8. No explanation on how the participants were recruited is provided.

The survey is organized by the National Health Commission. The Planning Department is responsible for coordination, and the National Statistical Information Center is responsible for the organization and implementation. They organize investigators to conduct unified training, investigation, data entry, quality control, supervision, etc. Shanghai Municipal Health Information Center provided us with the data in Shanghai. 

9. Example of the items that were develop for PJ and PG constructs should be provided.

According to your request, we have attached the questionnaire in the supplementary information.

10. Regarding the analyses performed major clarity and justification of the analyses performed should be presented (specially regarding the structural equation model).

Through literature review, we found that regression analysis and structural equation models are usually more commonly used analytical methods to verify the relationship between PE fit and attitude variables and related research hypothesis.

11. Descriptive statistics for the dependent and independent variables must the presented.

According to your advice, we had added a statistical description of the dependent and independent variables in the results section. (see the paragraph 1 of Results in red)

12.The discussion should not include results.

According to your advice, we had removed the results from the discussion.

13.The implications of the study should be further explored.

The relevant content has been clarified and supplemented in the part of implications. (See the Implications part in red)

14.The limitations of the study should include the limitations of self-reported questionnaires.

According to your advice, we added the content above into the limitations. (See the Limitations part in red)

In spite of our best efforts, may the shortcoming is still existed, but we think it could be good reference for the researchers and policy makers. Thank you so much for your valuable advice again.

Yours sincerely,

Guohong Li

School of Public Health, Shanghai Jiao Tong University School of Medicine, Shanghai, China

Center for HTA, China Hospital Development Institute, Shanghai Jiao Tong University, Shanghai, China

guohongli@sjtu.edu.cn

---

## [Decision Letter · Decision Letter 1]

30 Sep 2020

PONE-D-20-01262R1

How does person–environment fit affect Shanghai medical workers’ job satisfaction, turnover intention, and sense of accomplishment? An exploratory cross-sectional study

PLOS ONE

Dear Dr. Li,

Thank you for submitting your manuscript to PLOS ONE. After careful consideration, we feel that it has merit but does not fully meet PLOS ONE’s publication criteria as it currently stands. Therefore, we invite you to submit a revised version of the manuscript that addresses the points raised during the review process.

We look forward to receiving your revised manuscript.

Kind regards,

Geilson Lima Santana, M.D., Ph.D.

Academic Editor

PLOS ONE

Reviewers' comments:

Reviewer's Responses to Questions

**Comments to the Author**

1. If the authors have adequately addressed your comments raised in a previous round of review and you feel that this manuscript is now acceptable for publication, you may indicate that here to bypass the “Comments to the Author” section, enter your conflict of interest statement in the “Confidential to Editor” section, and submit your "Accept" recommendation.

Reviewer #2: (No Response)

2. Is the manuscript technically sound, and do the data support the conclusions?

Reviewer #2: Partly

3. Has the statistical analysis been performed appropriately and rigorously? 

Reviewer #2: Yes

4. Have the authors made all data underlying the findings in their manuscript fully available?

Reviewer #2: Yes

5. Is the manuscript presented in an intelligible fashion and written in standard English?

Reviewer #2: No

6. Review Comments to the Author

Reviewer #2: Review of the manuscript PONE-D-20-01262

How does person–environment fit affect Shanghai medical workers’ job satisfaction,

turnover intention, and sense of accomplishment? An exploratory cross-sectional study

Thank you for submitting the reviewed version of the paper. I do indeed see improvements in the methods and results section which respond both to the assessments of reviewer 1 and me. I also appreciate the elaboration of additional analysis and graphs to respond to some of my inquiries and allow me to conclude the data and results you are presenting are reliable and correct.

Nonetheless, there are several facts that you need to correct in order to make this a publishable article. The most important ones are related to the introduction and background.

1. Even though you restructured the introduction and part of the hypothesis section, the changes you made do not respond to my first inquire: the problem you are solving and its relevance is still not explicit in the document. Lines 68 and 93 make me think this is a replica or further extension of a previous study, however the results from that study are not mentioned in the paper, and the relevance of replicating or extending it are not present in the document. In plain language, why is this an article that some other researcher would want to read? Why is this article valuable? And What are your contributions to the literature? This should be explicit in the introduction, and the hypothesis section should be devoted to explain how you derive each hypothesis from a theory and previous data (the theoretical component is still missing)

2. Line 55, you need to first express what PE is (Person Environment fit?) before using an abbreviation (PE), this is valid also for line 59 PJ and PG.

3. Lines 62-65 are not clear, what do you want to express?

4. Lines 69-70 why did you choose those 2 dimensions, just because a different study selected other? You must have a better reason.

5. Lines 75-77, why is this relevant? What did this author conclude?

6. Line 78, what do you mean by progress? The following paragraph is confusing

7. Line 83-85, why is the concept of team important? Support with previous research and literature.

8. Line 90, what do you mean by saying that the value between individuals and groups is similar (complementary)?

9. Line 95, this is a different idea, you need to break the paragraph.

10. Lines 96- 103, you ignored my recommendation of not talking about personal accomplishment but professional efficacy, and elaborate an appropriate definition of the construct, however I didn’t find a satisfactory justification for this decision in your letter.

11. Lines 103-106, seem completely disconnected from the previous ideas. This needs further elaboration to allow the hypothesis to become a conclusion of what you have been saying.

12. Lines 110-120 explain results from previous research but it doesn’t provide an explanation or derivation for the following hypothesis.

13. Lines 133-135, I understand your aim is only to replicate the findings of Hassan. What is the reasoning for such relations to exist? And what are the implications or value of this study providing more evidence about it?

14. Line 152-153, you should remove the statement about prejudice or make explicit that because of the characteristics mentioned above it was minimized.

15. Lines 154- 167, now I really wonder if you were part of the survey collectors or if you received the collected information.

16. Line 172, what does it mean “considering workers in an organization”?

17. Line 295 you make assumptions of causality which you cannot make with your design. Avoid words such as affect.

18. Discussion line 13. Are you talking about you own results or results from different studies? You kept using words that imply causality such as effect.

19. Line 17 you talk again about effect

20. Line 19, why are you talking here about burnout if you did not develop these ideas in the introduction or background?

21. Lines 25-45, this is something new that you are contributing to the literature, you should spend more time developing this idea, and also introduce the relevance of those findings in the introduction… Also you should make explicit the analysis and results for this conclusion in the results sections even though there were not part of your hypothesis.

22. Lines 35-38 find some literature that help you support these conclusions.

23. Lines 57-63 review the implications… is this realist? Is it better to suggest modifications in selection processes to find doctors who have a better fit to this job?

24. Lines 65-67, is not clear how you get to that conclusion.

Overall I see you have made some progress, however you should work in greater depth on the introduction and trying to articulate the problem you are trying to solve. Even though PlosOne does not require a theoretical contribution, the author should make explicit what is the problem they are trying to solve with the study and what is the relevance of solving such a problem.

7. PLOS authors have the option to publish the peer review history of their article (what does this mean?). If published, this will include your full peer review and any attached files.

Reviewer #2: No

---

## [Author Response · Author response to Decision Letter 1]

14 Nov 2020

Dear editor Geilson Lima Santana,

Thank you for the opportunity to revise and resubmit the manuscript entitled "How does person–environment fit affect Shanghai medical workers’ job satisfaction, turnover intention, and sense of accomplishment? An exploratory cross-sectional study". We appreciate the insightful comments from the reviewer, and have made substantive revision base on the reviewer’ s advice. 

Response to the reviewer #2：

Thank you very much for your detailed instructions and suggestions. Based on the comments, we have revised our manuscript. And all the revisions are marked in red in the paper. Here we provide our explanation point by point to address your comments. 

1. Even though you restructured the introduction and part of the hypothesis section, the changes you made do not respond to my first inquire: the problem you are solving and its relevance is still not explicit in the document. Lines 68 and 93 make me think this is a replica or further extension of a previous study, however the results from that study are not mentioned in the paper, and the relevance of replicating or extending it are not present in the document. In plain language, why is this an article that some other researcher would want to read? Why is this article valuable? And What are your contributions to the literature? This should be explicit in the introduction, and the hypothesis section should be devoted to explain how you derive each hypothesis from a theory and previous data (the theoretical component is still missing)

Response: Thanks for the comments. We have revised the background section by adding research purposes and possible contributions to the literature. We have also summarized the results from previous research, and provided reasons for choosing PJ and PG fit as study variables in this study. 

This is study is an extension of a previous study. This study is unique for two reasons: (1) most PE fit studies have been in the Western context and in the industrial setting, and we extend the PE fit theory to the health setting in the Eastern context; and (2) This study focuses on medical staff in Shanghai, and examines PE fit and job outcomes in a high workload setting. We have added following sentences in the introduction section: “Most PE fit studies have been conducted in the Western context and in the industrial settings. In this study, we extend the PE fit theory to the health setting in China” and “Shanghai provides a good setting to examine factors, including PE fit, associated with job retention and satisfaction.” 

Additionally, we have added the following paragraph on the potential contribution of this study: “To our knowledge, this is the first PJ fit and PG fit study on medical staff at the city level in China. It adds existing literature on the impact of the PE fit on job outcome variables among health professionals that require strong team-work and under high pressure. Given prevalent issues of a shortage and high turnover of health professionals in developing countries, this study provides empirical evidence on the impact of PE fit on job-related outcomes.” 

We also appreciate your comments on hypotheses. We have revised them and provided more detailed explanation of them. In each research hypothesis, we have included the relevant theories. for example, we have applied to hypothesis 1 the input-process-output model (IPO), which shows that the process (e.g., teamwork, coordination) can have a positive impact on the team output (e.g., job satisfaction, team performance, and profession efficacy), to hypothesis 2 the Self-Verification Theory and Attraction Selection-Attrition model, and hypothesis 3 the conservation of resources (COR) theory. For detailed revision, please see our response to comment 12 and 13.

2. Line 55, you need to first express what PE is (Person Environment fit?) before using an abbreviation (PE), this is valid also for line 59 PJ and PG.

Response: Following your suggestion, we have spelt out “PE” before using it. The revised sentence reads as “Person–environment (PE) fit is broadly defined as the compatibility between the individual and their work environment, which occurs when their characteristics match well.2” We also spelt out PJ as person-job and PG as person-group. 

3. Lines 62-65 are not clear, what do you want to express?

Response: We have further clarified the sentences to strengthening the argument that PE assessment has been applied throughout individuals’ career cycle. The revised sentences read as “In the pre-hire stage, the measurement of PE fit was often used in career counseling, job searching and selection.5 At the same time, the majority of PE fit research has been conducted during the post hire period , showing strong linkage between single aspects of the PE fit and individual attitudes.6,7”

4. Lines 69-70 why did you choose those 2 dimensions, just because a different study selected other? You must have a better reason.

Response: We have provided additional reasons why only PJ and PG were selected for the study. There are two major reasons:

1. PJ and PG fit are the concepts more concentrated on team work, which is a main characteristics of medical practice in China. These two concepts are more pertinent to the analysis among medical staff. 

2. Compared to PO, PJ and PG fits are more modifiable in a relatively short period. 

We have added a paragraph to explain why PJ and PG were selected for the study. The added paragraph reads as: 

“Most PE fit studies have been conducted in the Western context and in the industrial settings. In this study, we extend the PE fit theory to the health setting in China. Expanding the understanding of PE fit among medical staff is important, because most physicians in China are working in teams in health facilities. 8,9 Among the four components of PE, PJ fit and PG fit are closely related subordinate concepts that concerns the alignment of workers and their job as a team. In this study, we focused on PJ and PG fit, because (1) team-based care models are dominant in clinical practice in China, demanding for strong team cooperation.10 Studying PJ fit and PG fit can help understand how they impact medical staff’s job attitude and behavior; and (2) we had previously explored the impact of PO fit on job satisfaction and turnover intention among community health workers in China and found that PO fit had a little association with turnover intention.11 We posited that the values of individuals and organization were relatively difficult to change in a short period of time12. Focusing on PJ and PG fits allow us to examine factors more modifiable to address attitude concerns among medical professionals, which helps improve the quality of medical care.13,14,15,16”

5. Lines 75-77, why is this relevant? What did this author conclude?

We aimed to provide the concepts of PJ fit and PG fit in the background section before getting into hypotheses. To make it clear for readers, we have added a subtitle of “the concept of PJ fit and PG fit”. We also revised the paragraph slightly to elaborate the three components in PJ fit. The revised paragraph now reads as “PJ fit is defined as the consistency or match between the characteristics of a person and the work or tasks performed.18 Edwards suggested two fundamental components of the demands-abilities fit and needs-supplies fit falling within PJ fit.1 When the employee’s knowledge, skills and abilities (KSA) match the demand of work, the fit of demands-abilities occurs; when the work performed meets the employee's need, the fit of needs-supplies occurs.19 Later, Scroggins20 added the concept of "self-concept of work fit" as an additional component to the PJ fit. Self-concept of work fit means that when job tasks match the individual’s self-concept, the individual will perceive the work as meaningful.20”

6. Line 78, what do you mean by progress? The following paragraph is confusing

We have revised the sentence to “Compared with PJ-fit research, there are much fewer studies on PG fit.22”. We also restructured and revised the whole paragraph. 

7. Line 83-85, why is the concept of team important? Support with previous research and literature.

This paragraph means to introduce the concept of PG. To avoid confusion, we removed the sentence. However, we explained previously that the medical practice in China is mostly team-based, thus we wanted to examine PG fit in this study. 

8. Line 90, what do you mean by saying that the value between individuals and groups is similar (complementary)?

We wanted to express that PG fit for the medical industry is mainly measured through value congruence, whereby the extent to which an individual shares similar values as their group members. To avoid confusion, we have removed the sentence.

9. Line 95, this is a different idea, you need to break the paragraph.

Following your advice, we have broken the paragraph.

10. Lines 96- 103, you ignored my recommendation of not talking about personal accomplishment but professional efficacy, and elaborate an appropriate definition of the construct, however I didn’t find a satisfactory justification for this decision in your letter.

Thank you for the suggestions. Following your suggestion, we have revised the whole paragraph and focused on the professional efficacy. The revised paragraph now reads as “According to the input-process-output model (IPO), the process experience (e.g., teamwork, coordination) is associated with individual and team outputs (e.g., job satisfaction, team performance, and profession efficacy) 24,25,26: coordination and encouragement from team members enhance individual efficacy, which optimize the collective group effort.26 Previous studies also showed a relationship between team work and job satisfaction.27,28, 29 Group settings can change individuals’ motivation,30,31 and group variables (e.g. group composition, leader and group characteristics) might affect individual members’ sense of fulfilment towards their work.32 Medical practice in China is mostly team-based, requiring coordination of health providers within the same department or across different department”

11. Lines 103-106, seem completely disconnected from the previous ideas. This needs further elaboration to allow the hypothesis to become a conclusion of what you have been saying.

We have revised the whole paragraph. See our response to comment 10. 

12. Lines 110-120 explain results from previous research but it doesn’t provide an explanation or derivation for the following hypothesis.

Following your suggestion, we have focused on the explanation of the hypothesis and revised the entire paragraph.

The revised paragraph reads as “Empirical studies have shown that PJ fit has an important impact on employees' work attitudes, such as job satisfaction and turnover intention.33 Smith34 found that job satisfaction was largely determined by employees’ work and their specific tasks. Caldwell’s study 35 suggested that satisfaction and performance were enhanced when individuals select a job that was compatible with his or her traits and skills. These findings were consistent with the self-verification theory36 and attraction selection-attrition model37 that predicted a negative relationship between meaningful work or the sense of job satisfaction and turnover intention. Meaningful work provides employees self-verified information, making the work itself attractive38,39. P-J fit has been established as an important predictor of job satisfaction and turnover intention, and we expect the same relationship will be held among medical staff in Shanghai.”

Hypothesis 2: Medical workers with higher PJ fit have: (a) higher job satisfaction and (b) lower turnover intention.”

13. Lines 133-135, I understand your aim is only to replicate the findings of Hassan. What is the reasoning for such relations to exist? And what are the implications or value of this study providing more evidence about it?

Following your suggestion previously about the hypothesis, we have revised the entire paragraph by explaining the hypothesis using prior studies or theories. One of reasons for examining such relationship is to identify potential mitigation factors to address the concern of job turnover. We have added following sentences in the revised paragraph: “Examining pathways on how PJ is associated with job turnover would help identify potential mitigation factors to address the concern of job turnover.” And the revised paragraph on hypothesis 3 now reads as following:

“Job satisfaction, reflecting the degree of personal enjoyment of work40, has shown its relevance to both individual job outcomes and organizational results, and it is a stable predictor of employee turnover intention and behavior.41,42, 43,44 Built on these findings and in combination with Hypothesis 2 that posits that PJ fit is associated with job satisfaction and job turnover directly, we further propose the third hypothesis of that job satisfaction could serves as a mediator between PJ fit and turnover intention, and PJ fit has both direct or indirect association with the turnover intention. In fact, Hassan M45 and Bindu Chhabra46 confirmed such relationship among employees in the Banking sector. Examining pathways on how PJ is associated with job turnover would help identify potential mitigation factors to address the concern of job turnover.”

14. Line 152-153, you should remove the statement about prejudice or make explicit that because of the characteristics mentioned above it was minimized.

Thanks for your suggestions, and we have removed the statement.

15. Lines 154- 167, now I really wonder if you were part of the survey collectors or if you received the collected information.

We did not participate in the survey, but we obtained the collected information and learned how the survey was conducted. We have clarified it by stating that it is National Health Commission of China that conducted the survey, and that we obtained the relevant NHSS data from Shanghai Health Commission and used the survey data from Shanghai only for this study. We have revised the paragraph, which reads as “We obtained the relevant NHSS data from Shanghai Health Commission and used the survey data from Shanghai only for this study. The survey includes all general hospitals and traditional Chinese medicine (TCM) general hospitals in 16 Shanghai districts. All tertiary hospitals were included in the survey; 50% secondary hospitals were randomly sampled and included in the survey; in each administrative district, 5 community health service centers were randomly selected. As to sampling health professionals, 10 clinicians and 5 nurses were randomly selected from each hospital, while 5 clinicians, 3 nurses, and 2 public health physicians were randomly selected from each community health service center. The survey contained a total of 2,600 health professionals. After eliminating missing values and logical errors from the samples, 2559 individuals were included in the analysis in this study.” 

16. Line 172, what does it mean “considering workers in an organization”?

We have revised it to “In terms of the type of health facilities where health professionals worked”

17. Line 295 you make assumptions of causality which you cannot make with your design. Avoid words such as affect.

Thanks for the suggestions, we have used association when discussing the results throughout the manuscript. 

18. Discussion line 13. Are you talking about you own results or results from different studies? You kept using words that imply causality such as effect.

Line 13 is the results from a study from someone else. We have removed the statistics obtained from the study to avoid the confusion. 

19. Line 17 you talk again about effect

We have corrected it, and it reads as “We also found that PG fit was directly associated with professional efficacy.”

20. Line 19, why are you talking here about burnout if you did not develop these ideas in the introduction or background?

We have removed burnout related sentence and revised the paragraph, which discusses more about the professional efficacy. The paragraph now reads as “We also found that PG fit was directly associated with professional efficacy. A study found that there was a strong correlation between PO fit and professional efficacy in nonmedical fields.52 Few studies have explored the analysis of professional efficacy as a variable in the field of PE. With the introduction of the concept of team-based care, the in-depth study of PG fit is particularly urgent. ”

21. Lines 25-45, this is something new that you are contributing to the literature, you should spend more time developing this idea, and also introduce the relevance of those findings in the introduction… Also you should make explicit the analysis and results for this conclusion in the results sections even though there were not part of your hypothesis.

Based on your suggestions, we have added one more objective in the background section. The added objective reads as “Another purpose of this study is to explore individual characteristics associated with PJ and PG fit.” 

In the method section, we have singled out the hierarchical regression analysis by making it as a separate paragraph. 

In the result section, we have added that a phrase of “to explore factors associated with PJ and PG fit” after the hierarchical regression analysis. 

22. Lines 35-38 find some literature that help you support these conclusions.

Thank you for your advice. We have added three references in the manuscript. These references are:

53.Boamah S A, Laschinger H. The influence of areas of work-life fit and work‐life interference on burnout and turnover intentions among new graduate nurses. Journal of Nursing Management, 2016, 24(2): E164-E174.

54. Sun WB. Briefly on the difficulties and countermeasures in the training of young doctors in my country at this stage. China Hospital, 2011, 15(05): 66-69

55. Jiang SL. Cultivation and improvement of doctor-patient communication skills of young cardiovascular surgeons. China Journal of Modern Medicine, 2016, 18(01): 92-93.

23. Lines 57-63 review the implications… is this realist? Is it better to suggest modifications in selection processes to find doctors who have a better fit to this job?

Thanks for the suggestion. We have modified the sentence to “The results of this study show that PJ fit is a critical factor determining job satisfaction and job turnover. It is important for hospital administrators to identify where the lack of fit lies between medical workers and work and to address it (e.g. through training), thereby reducing employee dissatisfaction and turnover intention. Research showed that turnover intention of medical staff was at a high level in China.56 Ensuring PE fit of health professionals might alleviate the persistent problems of labor shortage and frequent turnover of medical staff in China, which help improve the quality of medical care.”

Additionally, we have also added “In addition to the measurement of PJ fit when recruiting medical workers, it is also important to focus on strengthening the relevant fits among young physicians. It is possible to choose suitable position after rotation for individual employees by assessing the degree of fit between their job interests and the nature of their work. The medical professional association within hospitals or health facilities can also provide a communication platform for junior and senior physicians, which may be an effective way to improve their PJ fit as soon as possible.” 

24. Lines 65-67, is not clear how you get to that conclusion.

We have revised the sentence to “Given prevalent issues of a shortage and high turnover of health professionals in developing countries and the strong association between PJ fit and job turnover intention, it is important to deepen research on PJ fit in order to better address these chronic problems.”

We hope that we have thoroughly addressed the concerns that the reviewer has, and look forward to your decision. 

Yours sincerely,

Guohong Li

School of Public Health, Shanghai Jiao Tong University School of Medicine, Shanghai, China

Center for HTA, China Hospital Development Institute, Shanghai Jiao Tong University, Shanghai, China

guohongli@sjtu.edu.cn

---

## [Decision Letter · Decision Letter 2]

15 Dec 2020

PONE-D-20-01262R2

Person–environment fit and medical workers' job satisfaction, turnover intention, and professional efficacy: An exploratory cross-sectional study in Shanghai

PLOS ONE

Dear Dr. Li,

Thank you for submitting your manuscript to PLOS ONE. After careful consideration, we feel that it has merit but does not fully meet PLOS ONE’s publication criteria as it currently stands. Therefore, we invite you to submit a revised version of the manuscript that addresses the points raised during the review process.

We look forward to receiving your revised manuscript.

Kind regards,

Geilson Lima Santana, M.D., Ph.D.

Academic Editor

PLOS ONE

Reviewers' comments:

Reviewer's Responses to Questions

**Comments to the Author**

1. If the authors have adequately addressed your comments raised in a previous round of review and you feel that this manuscript is now acceptable for publication, you may indicate that here to bypass the “Comments to the Author” section, enter your conflict of interest statement in the “Confidential to Editor” section, and submit your "Accept" recommendation.

Reviewer #2: All comments have been addressed

2. Is the manuscript technically sound, and do the data support the conclusions?

Reviewer #2: Yes

3. Has the statistical analysis been performed appropriately and rigorously? 

Reviewer #2: Yes

4. Have the authors made all data underlying the findings in their manuscript fully available?

Reviewer #2: Yes

5. Is the manuscript presented in an intelligible fashion and written in standard English?

Reviewer #2: Yes

6. Review Comments to the Author

Reviewer #2: Review of the manuscript PONE-D-20-01262

How does person–environment fit affect Shanghai medical workers’ job satisfaction,

turnover intention, and sense of accomplishment? An exploratory cross-sectional study

Thank you for submitting the second reviewed version of the paper. In this opportunity I am able to see significant improvement on the manuscript, especially in the introduction and theory sections. I consider your manuscript is almost ready to be published and thus I suggest some minor revisions to it. The most relevant issues are:

1. In the introduction you indeed mention that it is relevant to expand understanding on PE fit (lines 71-72) because most physician in the country are working in teams. However, you need to create closure of the idea you want to convey and be able to directly connect this fact (many doctors working in groups) with your theory. Also when you express the idea in that way you are presuming that in the western countries physicians work in isolation, and I am not sure that is the case (review the work of Gittell on Relational coordination).

2. You need to explain the relevance of your variable of interest before you propose the hypothesis, thus you need to explain before what those variables are and why it is important to find a way to improve them.

3. Review the limitations section, especially the last idea you present as it is incomplete.

4. Make a thorough revision of writing, spelling and grammar, as some sentences are not clear (lines 27, 31, 38, 65, review line 287 when to talk about job achievement).

7. PLOS authors have the option to publish the peer review history of their article (what does this mean?). If published, this will include your full peer review and any attached files.

Reviewer #2: **Yes: **Maria Camila Umana

---

## [Author Response · Author response to Decision Letter 2]

29 Dec 2020

Dear editor Geilson Lima Santana,

Thank you for the opportunity to revise and resubmit the manuscript entitled "Person–environment fit and medical workers' job satisfaction, turnover intention, and professional efficacy: An exploratory cross-sectional study in Shanghai". We appreciate the insightful comments from the reviewer, and have made substantive revision base on the reviewer’s comments. 

Response to the reviewer #2：

Thank you very much for your detailed instructions and suggestions. Based on the comments, we have revised our manuscript. And all the revisions are marked in red in the paper. Here we provide our explanation point by point to address your comments. 

1. In the introduction you indeed mention that it is relevant to expand understanding on PE fit (lines 71-72) because most physician in the country are working in teams. However, you need to create closure of the idea you want to convey and be able to directly connect this fact (many doctors working in groups) with your theory. Also when you express the idea in that way you are presuming that in the western countries physicians work in isolation, and I am not sure that is the case (review the work of Gittell on Relational coordination). 

Response: Thanks for the comments. Following your suggestion, we deleted the sentence and elaborated on the theory of relational coordination. The revised sentence reads as “Similar to the medical practice in Western countries, the medical practice in China often operates in teams and thus relational coordination among medical professionals directly impacts on the quality of care.” We also cited Gittell’s work to strengthen the argument. 

2. You need to explain the relevance of your variable of interest before you propose the hypothesis, thus you need to explain before what those variables are and why it is important to find a way to improve them.

Response: Thanks for the suggestion. We have restructured the introduction section substantially, and moved the definition of PJ and PG fit up to highlight these two variables as key independent variables, because they are more modifiable compared to PO fit. We also move the justification of using Shanghai as the study setting given the high workload among health professionals, and to highlight the three job outcome variables as main dependent variables. 

3. Review the limitations section, especially the last idea you present as it is incomplete.

Response: We have revised the limitation section substantially. For example, the third limitation now reads as “Third, we used self-reported questionnaires (tools) to measure key research variables in this study, and the reported results may be subjective to potential bias. For example, although we have tested the construct validity of questionnaires, participants can still provide untruthful responses, and it is not clear if the questioners used in this study have a good criteria validity to capture the essence of what we would like to measure. Future research should validate the questionnaires from other perspectives (e.g. criteria validity and content validity).”

4. Make a thorough revision of writing, spelling and grammar, as some sentences are not clear (lines 27, 31, 38, 65, review line 287 when to talk about job achievement).

Response: Thank you for your suggestion. We have made substantive editing: restructuring the introduction section; correcting grammar and spelling mistakes, and clarifying sentences.

We hope that we have thoroughly addressed the concerns that the reviewer has, and look forward to your decision. 

Yours sincerely,

Guohong Li

School of Public Health, Shanghai Jiao Tong University School of Medicine, Shanghai, China

Center for HTA, China Hospital Development Institute, Shanghai Jiao Tong University, Shanghai, China

guohongli@sjtu.edu.cn

---

## [Decision Letter · Decision Letter 3]

9 Feb 2021

PONE-D-20-01262R3

Person–environment fit and medical professionals’ job satisfaction, turnover intention, and professional efficacy: A cross-sectional study in Shanghai

PLOS ONE

Dear Dr. Li,

Thank you for submitting your manuscript to PLOS ONE. After careful consideration, we feel that it has merit but does not fully meet PLOS ONE’s publication criteria as it currently stands. Therefore, we invite you to submit a revised version of the manuscript that addresses the points raised during the review process.

We look forward to receiving your revised manuscript.

Kind regards,

Geilson Lima Santana, M.D., Ph.D.

Academic Editor

PLOS ONE

Reviewers' comments:

Reviewer's Responses to Questions

**Comments to the Author**

1. If the authors have adequately addressed your comments raised in a previous round of review and you feel that this manuscript is now acceptable for publication, you may indicate that here to bypass the “Comments to the Author” section, enter your conflict of interest statement in the “Confidential to Editor” section, and submit your "Accept" recommendation.

Reviewer #2: All comments have been addressed

2. Is the manuscript technically sound, and do the data support the conclusions?

Reviewer #2: Yes

3. Has the statistical analysis been performed appropriately and rigorously? 

Reviewer #2: Yes

4. Have the authors made all data underlying the findings in their manuscript fully available?

Reviewer #2: Yes

5. Is the manuscript presented in an intelligible fashion and written in standard English?

Reviewer #2: Yes

6. Review Comments to the Author

Reviewer #2: Thank you for submitting a new version of the manuscript. After careful revision of the I can see an improvement specifically in the issues that I mentioned in my last review. I also consider the manuscript is now coherent and clear about the theory, the aim, the design of the study and its conclusions. However, in this opportunity I find one major issue you need to carefully address before publishing this article and several other issues you need to check.

Regarding themajor issue, please clarify the ethical implications analysis you made and report it in the corresponding section. Did the participants gave their consent to collect the data and did they gave permission to use it in research studies? What were the risks of participating in such study? Even if it does not represent a risk, you should make a statement about it. What does it mean that the study did “not require ethical approval”?... if the institutions that support this study does not require and IRB process, you should at least show a reflection about the ethical implications of this study for the participants, even if they are not patients or public, participants are human beings who must decide if they want to participate in studies or not. I kindly suggest that you review if the National Health Services Survey has made an analysis of the ethical implications of collecting the data and allowing to use it for research and if they collect informed consents from participants.

Regarding the minor issues, I provide a detailed report below. The line number refer to the tracked changes version of the manuscript.

101. “Later, Scroggins....” indicates that you are explaining the theory in a chronological way and you are not, simply avoid the later.

106-108. Two sentences explaining the same idea, please unify.

109-110 The collaborative nature of medical practice is a good argument for studying PG fit, but not PJ fit. What is the argument for studying PJ fit?

115-116 Review the way numbers are presented. Separate average number of patients from bed occupation.

117-122 You present three different ideas without a context, or which are not connected to the rest of the paragraph: “Medical professionals have a high workload” and is a phrase out of context with the rest of the paragraph. “Maintaining a high level of job satisfaction has been challenging” is not connected with the rest of the ideas of the paragraph. “In this study we have not intended to assess the impact of PO fit” needs to be better connected with the following idea and makes part of a separate paragraph.

122-129 Please reformulate to make clear.

176-180 It is not necessary to make a comparison with western countries, only to refer to the idea that working in coordination with others is central for the health care setting (regardless of the location). One way to do this is to use the concepts of relational coordination, but there might be other similar concepts or literature that refers to this notion.

190-191 If you refer to those theories you should briefly explain what they say in general, why are you using two theoretical models to explain the same thing? What are the differences between those theories?

194-195 Please provide the cites and references for the studies that have established such relations.

234 What does it mean “ilogical errors from the sample”?

265 Separate the measures of PJ and PG. You use the subtitle PE fit, but you are measuring two of the three components of PE fit, and theorizing about them separately.

267-279 How did you create these items? Are they adapted from a different survey or instrument? Please describe where does it come from and provide examples of the items.

280-284 Check grammar and spelling.

291 –293 You refer to the components, but how many items did this scale included?

301 Be consistent in the writing... the previous paragraphs refer in passive voice to how the constructs were measured and in this paragraph and the next one you introduce the NHSS

308-323 You should only inlcude one sample item, by including all of them you are violating the terms and conditions of use of the MBI which should have been purchased.

316-317 An alpha of .84 is often considered good.

329 You can eliminate the “calculate Cronbach coefficient to check internal consistency as this is what a Cronbach coefficient does.

352-357 Please give some context on how to interpretate the values of the described variables... are they high, low, average compared to other samples using the same measures?

396. A mediating factor between what and what?

398 Conclude the idea by specifying if the hypothesis was supported or rejected.

Discussion

2 Please start the discussion by posing again the main objective of the study. Also, in the discussion you should not use the specific results and statistics. Please remember that the discussion summarizes the main results by explaining its conceptual meaning. Because those statistics are already clear in the results section you should eliminate them from the discussion.

In the discussion you should be clear about the theoretical or conceptual implications of your findings.

35-36 You talk about improving PJ fit of younger physicians. How could that be done?

92-95 How would a communication platform help to improve PJ fit?

115 As you did not conduct a study with an intervention, your study does not lead to the conclusion that strengthening PG fit could improve professional efficacy. Maybe your study allows to think that, but is not a factual conclusion. Please reformulate.

7. PLOS authors have the option to publish the peer review history of their article (what does this mean?). If published, this will include your full peer review and any attached files.

Reviewer #2: No

---

## [Author Response · Author response to Decision Letter 3]

2 Mar 2021

Dear editor Geilson Lima Santana,

Thank you for the opportunity to revise and resubmit the manuscript entitled "Person–environment fit and medical workers' job satisfaction, turnover intention, and professional efficacy: An exploratory cross-sectional study in Shanghai". We appreciate the insightful comments from the reviewer, and have made substantive revision base on the reviewer’s comments. 

Response to the reviewer #2：

Thank you very much for your detailed instructions and suggestions. Based on the comments, we have revised our manuscript. And all the revisions are marked in red in the paper. Here we provide our explanation point by point to address your comments. 

1. Regarding the major issue, please clarify the ethical implications analysis you made and report it in the corresponding section. Did the participants give their consent to collect the data and did they give permission to use it in research studies? What were the risks of participating in such study? Even if it does not represent a risk, you should make a statement about it. What does it mean that the study did “not require ethical approval”?... if the institutions that support this study does not require and IRB process, you should at least show a reflection about the ethical implications of this study for the participants, even if they are not patients or public, participants are human beings who must decide if they want to participate in studies or not. I kindly suggest that you review if the National Health Services Survey has made an analysis of the ethical implications of collecting the data and allowing to use it for research and if they collect informed consents from participants.

Response: Thanks for the suggestion. What we want to express is that this study was based on the secondary analysis of existing National Health Services Survey datasets. The National Health Service Survey began in 1993 and is conducted every five years. It was organized by the National Health Commission, the Planning Department is responsible for coordination, and the Regional Health Information Center is responsible for implementation. All data in this study were obtained and available for research upon the approval from authorized Shanghai Municipal Health Information Center. The data in this study did not contain the identifiable private information. 

The participants of original National Health Services Survey were adequately informed about all relevant aspects of the survey, including its objective and procedures. The survey had obtained their informed consent. All participants had the right to decide whether to participate in the survey, and could decide whether to withdraw at any time.

2. 101. “Later, Scroggins....” indicates that you are explaining the theory in a chronological way and you are not, simply avoid the later.

Response: Following your suggestion, we deleted the conjunction.

3. 106-108. Two sentences explaining the same idea, please unify. 

Response: We have revised these sentences. The revised sentence reads as “Compared with PJ-fit research, there is little research on antecedents of PG fit, and how it affects the performance of the team to which an individual belongs [13,14].”

4. 109-110 The collaborative nature of medical practice is a good argument for studying PG fit, but not PJ fit. What is the argument for studying PJ fit? 

Response: Following your suggestion, we have added relevant content. The revised sentence reads as “Given the collaborative nature of the medical practice and the professionalism and particularity of the work itself [15], we focused our analysis on the potential effect of PJ and PG fit on job outcomes, particularly on job satisfaction and employees’ intent to quit in Shanghai. Previous studies show that PJ and PG fit has a strong correlation with job satisfaction [1, 16].”

5. 115-116 Review the way numbers are presented. Separate average number of patients from bed occupation. 

Response: Following your suggestion, we have separated average number of patients from bed occupation. The revised sentence reads as “The reports showed that the average number of patients that a physician treats per day were 14.4 in Shanghai and hospital bed utilization were 95.85%.”

6. 117-122 You present three different ideas without a context, or which are not connected to the rest of the paragraph: “Medical professionals have a high workload” and is a phrase out of context with the rest of the paragraph. “Maintaining a high level of job satisfaction has been challenging” is not connected with the rest of the ideas of the paragraph. “In this study we have not intended to assess the impact of PO fit” needs to be better connected with the following idea and makes part of a separate paragraph. 

Response: Thanks for the suggestion. We have restructured this paragraph and moved the sentence of “Maintaining a high level of job satisfaction has been challenging.” We also have separated the rest of the ideas into next paragraph.

7. 122-129 Please reformulate to make clear. 

Response: Thanks for the suggestion. We have reformulated this paragraph. The revised content reads as “In this study, we have not intended to assess the impact of PO fit. We had previously explored the impact of PO fit on job satisfaction and turnover intention among community health workers in China and found little impact of PO fit on the turnover intention [19]. It was also found that the difference in the PO fit of the survey subjects was small, which may be related to that they were all from public medical institutions. [19] What’s more the values of individuals and organizations might be relatively difficult to change in a short time [20]. On the contrast, the PJ fit and PG fit are more modifiable to address attitude concerns among medical professionals, so as to improve the quality of medical care [21-24].”

8. 176-180 It is not necessary to make a comparison with western countries, only to refer to the idea that working in coordination with others is central for the health care setting (regardless of the location). One way to do this is to use the concepts of relational coordination, but there might be other similar concepts or literature that refers to this notion. 

Response: Following your suggestion, we propose research hypotheses based on the concept of relationship coordination theory: “Based on the theory of relationship coordination, it is also a form of organizing social capital, which can make it easier for people to obtain resources needed to accomplish one's work.[34] We know that having the resources needed to accomplish the work has been proved to be an important source of job satisfaction. [35] Medical practice is mostly team-based, requiring coordination of health providers within the same department or across different departments.”

9. 190-191 If you refer to those theories you should briefly explain what they say in general, why are you using two theoretical models to explain the same thing? What are the differences between those theories? 

Response: Following your suggestion, we respectively introduced the specific content of these two theoretical models. The theory of self-verification showed that “self-consistency improves the degree to which the individual feel that he can control and manipulate his surrounding environment. A stable self-concept allows individuals to negotiate social reality and understand how to act effectively in a given situation.” And the attraction–selection–attrition model told us that “individuals will be attracted to and seek out jobs and organizations that provide them with meaningful self-confirming information and will likely continue in the job as long as self-confirming information is received and a high level of self-concept–job fit is perceived.”

10. 194-195 Please provide the cites and references for the studies that have established such relations. 

Response: Following your suggestion, we have added references as follows.

Tinsley HEA. The congruence myth: An analysis of the efficacy of the person–environment fit model. Journal of Vocational Behavior. 2000;56(2):147-79.

11. 234 What does it mean “ilogical errors from the sample”? 

Response: During the process of data cleaning, we found some logic errors. For example, there are two identical questions in the questionnaire to test whether the participants answered earnestly. If there was an inconsistency, the sample was regarded as a logical error.

12. 265 Separate the measures of PJ and PG. You use the subtitle PE fit, but you are measuring two of the three components of PE fit, and theorizing about them separately. 

Response: Following your suggestion, we have revised the subtitle into PJ fit and PG fit. 

13. 267-279 How did you create these items? Are they adapted from a different survey or instrument? Please describe where does it come from and provide examples of the items. 

Response: Following your suggestion, we have added the process of revision and given a sample item of “My personality is a good match for this job.” 

14. 280-284 Check grammar and spelling. 

Response: Thanks for the suggestion. We have revised this sentence. The revised content reads as “The PG fit was measured using the scale constructed by Piasentin and Chapman (2007), and revised for medical workers. It included three aspects matching an individual and the team that the individual belonged to and was a 4-point Likert scale (1 = completely disagree; 4 = fully agree).”

15. 291 –293 You refer to the components, but how many items did this scale included? 

Response: Thanks for the suggestion. This scale has 5 items. We have corrected in the manuscript.

16. 301 Be consistent in the writing... the previous paragraphs refer in passive voice to how the constructs were measured and in this paragraph and the next one you introduce the NHSS 

Response: Thanks for the suggestion. We have revised the sentence to keep consistent.

17. 308-323 You should only inlcude one sample item, by including all of them you are violating the terms and conditions of use of the MBI which should have been purchased. 

Response: Thanks for the suggestion. We have deleted the whole items and kept a sample item in each scale.

18. 316-317 An alpha of .84 is often considered good. 

Response: Thanks for the suggestion. The revised content reads as “The Cronbach’s � was 0.840.”

19. 329 You can eliminate the “calculate Cronbach coefficient to check internal consistency as this is what a Cronbach coefficient does. 

Response: Following your suggestion, we have removed these words.

20. 352-357 Please give some context on how to interpretate the values of the described variables... are they high, low, average compared to other samples using the same measures? 

Response: Thanks for the suggestion. Initially, we hope to compare these results between related studies in the discussion section. However, we found that some studies with the same measures have not presented these values, such as reference [1,2]. 

Some studies that presented these values showed that the degree of job satisfaction among teachers was 3.87 [3] and among employees was 3.61 [4]. The turnover intention of private sector employee was 3.16.[5] These studies were carried out in different countries and different industries. Due to the particularity of the sample, relatively few researches have been carried out in the medical field. This also increased the difficulty of comparison between the same measurement tools.

Our research had a reasonable sampling design, which made the results more representative. It can be said that this was the first PE fit study at the city level, and our results can reflect the overall situation of medical staff in Shanghai. We still retained this content in the results section, because we hope to let readers understand the distribution of our samples in these scales, and to provide a basis for follow-up research to compare with.

[1] Scroggins W A. Antecedents and outcomes of experienced meaningful work: A person-job fit perspective. The Journal of Business Inquiry, 2008, 7(1): 68-78.

[2] Wardana M C, Anindita R, Indrawati R. Work Life Balance, Turnover Intention, And Organizational Commitment in Nursing Employees at X Hospital, Tangerang, Indonesia. Journal of Multidisciplinary Academic, 2020, 4(4): 221-228.

[3] Ho C L, Au W T. Teaching satisfaction scale: Measuring job satisfaction of teachers. Educational and psychological Measurement, 2006, 66(1): 172-185.

[4] Moorman R H. The influence of cognitive and affective based job satisfaction measures on the relationship between satisfaction and organizational citizenship behavior. Human relations, 1993, 46(6): 759-776.

[5] Yin-Fah B C, Foon Y S, Chee-Leong L, et al. An exploratory study on turnover intention among private sector employees. International Journal of Business and Management, 2010, 5(8): 57.

21. 396. A mediating factor between what and what? 

Response: Thanks for the suggestion. The revised sentence reads as “To validate Hypothesis 3, we constructed a pathway using job satisfaction as a mediating factor between PJ fit and turnover intention.”

22. 398 Conclude the idea by specifying if the hypothesis was supported or rejected. 

Response: Thanks for the suggestion. The revised sentence reads as “We found that PJ fit was indirectly associated with job turnover intention, and the association was statistically significant (�=−0.185, P<0.001), which supported Hypothesis 3.”

23. 2 Please start the discussion by posing again the main objective of the study. Also, in the discussion you should not use the specific results and statistics. Please remember that the discussion summarizes the main results by explaining its conceptual meaning. Because those statistics are already clear in the results section you should eliminate them from the discussion. 

Response: Following your advice, we have added the paragraph of the main objective of the study and deleted the specific results and statistics from the discussion. The revised paragraph reads as “In keeping with the PE fit theory, we introduce a theoretical model that present hypotheses about the relations of PJ and PG fits with job satisfaction, turnover intention, and professional efficacy. Additionally, we explored the mediation mechanism of job satisfaction in PJ fit on turnover intention.”

24. In the discussion you should be clear about the theoretical or conceptual implications of your findings. 

Response: We have revised the whole paragraph. See our response to comment 23.

25. 35-36 You talk about improving PJ fit of younger physicians. How could that be done? 

Response: We have put forward practices and suggestions for reference in the implications section. The suggestion reads as “In addition to the measurement of PJ fit when recruiting medical workers, it is also important to focus on strengthening the relevant fits among young physicians. It is possible to choose a suitable position after rotation for individual employees by assessing the degree of fit between their job interests and the nature of their work. The medical professional association within hospitals or health facilities can provide a communication platform for junior and senior physicians, where senior physicians can share their work experience and self-improvement methods with junior physicians through pairing or group sharing to improve their PJ fit.”

26. 92-95 How would a communication platform help to improve PJ fit? 

Response: Thanks for the suggestion. We have added the explanation. The revised sentence reads as “The medical professional association within hospitals or health facilities can provide a communication platform such as social media for junior and senior physicians, where senior physicians can share tacit knowledge according to the physicians’ perspectives and experiences [58]and self-improvement methods with junior physicians through pairing or group sharing to improve their PJ fit.”

27. 115 As you did not conduct a study with an intervention, your study does not lead to the conclusion that strengthening PG fit could improve professional efficacy. Maybe your study allows to think that, but is not a factual conclusion. Please reformulate. 

Response: Thanks for the suggestion. The revised sentence reads as “The results show that medical workers with higher PJ or PG fits have higher job satisfaction and those with higher PG fit have higher professional efficacy.”

We hope that we have thoroughly addressed the concerns that the reviewer has, and look forward to your decision. 

Yours sincerely,

Guohong Li

School of Public Health, Shanghai Jiao Tong University School of Medicine, Shanghai, China

Center for HTA, China Hospital Development Institute, Shanghai Jiao Tong University, Shanghai, China

guohongli@sjtu.edu.cn

---

## [Decision Letter · Decision Letter 4]

13 Apr 2021

Person–environment fit and medical professionals’ job satisfaction, turnover intention, and professional efficacy: A cross-sectional study in Shanghai

PONE-D-20-01262R4

Dear Dr. Li,

We’re pleased to inform you that your manuscript has been judged scientifically suitable for publication and will be formally accepted for publication once it meets all outstanding technical requirements.

Kind regards,

Geilson Lima Santana, M.D., Ph.D.

Academic Editor

PLOS ONE

Additional Editor Comments (optional):

Reviewers' comments:

Reviewer's Responses to Questions

**Comments to the Author**

1. If the authors have adequately addressed your comments raised in a previous round of review and you feel that this manuscript is now acceptable for publication, you may indicate that here to bypass the “Comments to the Author” section, enter your conflict of interest statement in the “Confidential to Editor” section, and submit your "Accept" recommendation.

Reviewer #2: All comments have been addressed

2. Is the manuscript technically sound, and do the data support the conclusions?

Reviewer #2: Yes

3. Has the statistical analysis been performed appropriately and rigorously? 

Reviewer #2: Yes

4. Have the authors made all data underlying the findings in their manuscript fully available?

Reviewer #2: Yes

5. Is the manuscript presented in an intelligible fashion and written in standard English?

Reviewer #2: Yes

6. Review Comments to the Author

Reviewer #2: Thanks for addresssing the comments I have made in the past. I consider the paper is sound and presents original results from research that will enrich the literature.

7. PLOS authors have the option to publish the peer review history of their article (what does this mean?). If published, this will include your full peer review and any attached files.

Reviewer #2: No

---

## [Editor Report · Acceptance letter]

16 Apr 2021

PONE-D-20-01262R4 

Person–environment fit and medical professionals’ job satisfaction, turnover intention, and professional efficacy: A cross-sectional study in Shanghai 

Dear Dr. Li:

I'm pleased to inform you that your manuscript has been deemed suitable for publication in PLOS ONE. Congratulations! Your manuscript is now with our production department. 

Kind regards, 

on behalf of

Dr. Geilson Lima Santana 

Academic Editor

PLOS ONE